

# Skating on slippery ice

**J. M. J. van Leeuwen**

Instituut-Lorentz, Universiteit Leiden,
Niels Bohrweg 2, 2333 CA Leiden, The Netherlands

## Abstract

The friction of a stationary moving skate on smooth ice is investigated, in particular in relation to the formation of a thin layer of water between skate and ice. It is found that the combination of ploughing and melting gives a friction force that is rather insensitive for parameters such as velocity and temperature. The weak dependence originates from the pressure adjustment inside the water layer. For instance, higher velocities, giving rise to higher friction, also lead to larger pressures, which, in turn, decrease the contact zone and so lower the friction. By treating ice as a Bingham solid the theory combines and completes two existing but conflicting theories on the formation of the water layer.

doi:[10.21468/SciPostPhys.3.6.042](10.21468/SciPostPhys.3.6.042)


## 1  Introduction

Ice seems to be the only substance on which one can conveniently skate, which prompts the question: "what sort of special properties does ice have as compared to other solids?" Moreover one can glide on ice over a wide range of velocities, types of skates and temperatures. Ice is in many respects a peculiar solid and there is much folklore about the mystery of skating.

Ice is one of the few substances where the solid is less dense than the liquid, which has a profound impact in nature. Skating is a minor beneficiary of this property, as canals freeze on top, so one does not have to wait till the canal is solidly frozen. Another interesting property of water is that the melting line in the pressure-temperature plane has an unusual slope: with increasing pressure the freezing temperature lowers, while usually pressure favours the solid phase. It is illustrated in the famous high-school experiment where a steel cable with weights on both sides, melts itself through a block of ice at temperatures below zero, such that the block refreezes on top of the steel cable! This property has featured for quite a while as explanation for skating: due to the pressure exerted by the skater on the ice, a water layer forms and the skates glide on this water layer. It has been demonstrated several times that this explanation is not feasible [1,2]. Although the lowering of the melting point under pressure does not explain the skating phenomenon, its influence can not be dismissed at low temperatures, as we will show.

The slipperiness of ice has also been attributed to the special structure of the free ice surface. The existence of a water layer on the surface, even without skating, was already suggested by Faraday [3]. Computations and measurements indicate that this layer is only a few molecules thick, such that one cannot speak of this water layer as a hydrodynamic system, see e.g. [4]. For slow velocities and low temperatures the structure of the surface plays an important role on the friction properties [5].

In this paper we study the formation and influence of the water layer underneath the skate for usual conditions, i.e for sliding velocities of meters per second and temperatures of a few degrees below the melting point of ice. Gliding is only a part of the physics of skating. Also important is the ability to push oneself forward, which is possible due to the shape of the skate and to the fact that ice is easily deformable.

The main argument for the formation of a water layer, is that friction generates heat and that heat melts ice. How much of the heat melts ice and which part leaks away, is an important issue, which we address in this paper. We will treat the water layer as a hydrodynamic system, which implies that its thickness has to be at least of the order of 10 nm. If such a layer of

water is formed, the hydrodynamic properties of the layer determine the friction, which then becomes independent of the surface properties.

The physics of the water layer between skate and ice is not simple, with a rich history, see e.g. [2,4,6]. In spite of the fact that the problem is century old, the water layer has never been directly observed. A potential method for observation is based on the difference in dielectric properties of ice and water at high frequencies [7]. Indirect evidence for the water layer may result from measurements of the friction of a skate on ice. If friction is mediated through a water layer, then its characteristics can be checked. This paper deals with a calculation of the friction.

It is well known that a skater on virgin ice leaves a trail. Is this trail due to melting or to plastic deformation (ploughing)? The deformation is plastic if the exerted pressure exceeds the hardness of ice. The trail is an indication that the deformation of ice is plastic. Indeed, the weight of a skater of, say 72 kg, cannot be supported by an elastic deformation of ice. Moreover skates have sharp edges which will make kinks in the surface of ice (even in horizontal position) and near a kink the pressure will always exceed the hardness of ice. Therefore we focus on plastic deformation of the ice and justify this *a posteriori* by the high pressures occurring in the water layer for skating speeds.

At the moment there are two quantitative but competing theories for the formation of a water layer and the furrow of the trail. The one by Lozowski and Szilder [8], assumes that most of the dent in the ice is the result of ploughing. The other theory, by Le Berre and Pomeau [9] assumes that the dent is due to melting only. We will show which fraction of the trail is due to melting and which is due to ploughing. The two regimes, melting and ploughing merge continuously. Although our description is a unification of both theories, the results are substantially different from both theories.

In this paper we discuss the issue in the simplest possible setting: a speed skater moving in upright position over the ice with a velocity $V$ on perfectly smooth ice and skates. The skater stands with his mass $M$ on one skate. For skating near the melting point of ice, heat flows into the skates and into the ice are less important and we discuss their influence later on. Our main concern is the thickness of the water layer *underneath* the skate; the water films that form at the *sides* of the skate, play a minor role.

The only measurements of the friction of skates under realistic conditions, that we are aware of, have been performed by de Koning et al. [2]. Their skater had a velocity of speed of $V = 8$ m/s and a weight of 72 kg. Together with the standard parameters of skates: curvature $R = 22$ m and width $w = 1.1$ mm, we call these specifications the skating conditions. Unless otherwise stated, our calculations are carried out for temperature $T = 0^0$C. We take the skating conditions as reference point and vary the parameters individually with respect to this point.

The various aspects of the theory are presented in Sections in the following order:

**2** describes the used coordinate systems and the geometry of the skate.

**3** provides the necessary information on the material constants of water and ice.

**4** gives the force balance for a static skater.

**5** derives the heat balance, determining the thickness of the water layer.

**6** solves the equations for the thickness of the water layer in the regime where only melting plays a role.

**7** summarises the necessary formulas for hydrodynamics and pressure of the water layer.

**8** yields the shape of the water layer in the ploughing regime.

**9** treats the cross-over from the ploughing to the melting regime.

**10** calculates the pressure in both regimes.

**11** relates the weight of the skater to the pressure in the water layer. Also the slowing down force of the ice is computed, which is the sum of the friction in the water layer and the ploughing force.

**12** contains the velocity dependence of the friction.

**13** discusses the influence of the ice temperature on the friction.

**14** closes the paper with a discussion of the approximations and a comparison with the existing theories.

In addition a number of separate issues are treated in Appendices.

## 2 Geometry of the skates

For the description of the phenomena we need two coordinate systems: the ice fixed system and that of the moving skater. If $x, y, z$ are the coordinates in the ice system, then the coordinates $x', y', z'$ of the same point in the skate system are

$$x' = x - Vt, \qquad y' = y, \qquad z' = z, \tag{1}$$

where V is the velocity of the skate. The $x$ coordinate points in the forward direction of the skate. The origin of the skate coordinates is in the middle of the skate at the level of the ice. The lowest point of the skate, the depth of the trail, is a distance $d$ below the original ice level. The $y$ direction is horizontally and perpendicular to the skate blade and the $z$ direction points downward into the ice. At time $t = 0$ the two coordinate systems coincide. See Fig. 1 for a cross-section in the longitudinal direction. $d(x')$ is the locus of the bottom of the skate. With $R$ the curvature radius of the skate it is given by the equation

$$[R - d + d(x')]^2 + x'^2 = R^2, \qquad \text{or} \qquad d(x') = [R^2 - x'^2]^{1/2} + d - R. \tag{2}$$

In the ice system we have correspondingly

$$d(x, t) = d(x') = [R^2 - (x - Vt)^2]^{1/2} + d - R. \tag{3}$$

So for a fixed point $x$ in the ice system, the downward velocity of the skate $v_{sk}(x)$ is at $t = 0$

$$v_{sk}(x) = \left( \frac{\partial d(x, t)}{\partial t} \right)_{t=0} = V \frac{x}{[R^2 - x^2]^{1/2}} \simeq V \frac{x}{R}. \tag{4}$$

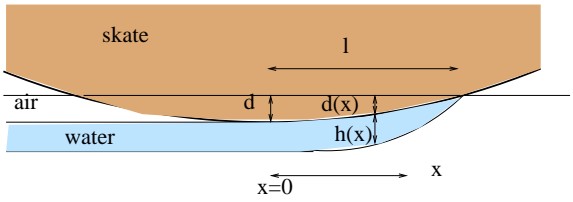

Figure 1: longitudinal cross-section of skate, ice and water layer in between.

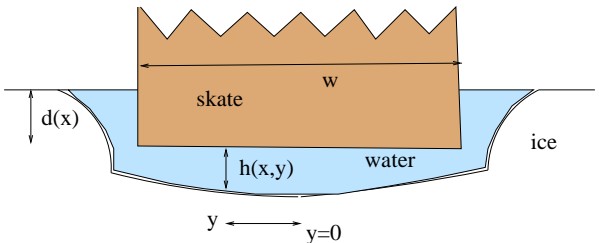

Figure 2: transverse cross-section in the $y,z$ plane of the skate and the layer of water underneath. Note that, for visibility, width (mm) and depth ($\mu$m) are not drawn in proper scale.

| material constant | symbol | value | unit |
|---|---|---|---|
| dynamic viscosity water | $\eta$ | $1.737*10^{-3}$ | Pa s |
| thermal conductivity water | $\kappa_{\mathrm{w}}$ | 0.591 | J/(m s K) |
| thermal conductivity ice | $\kappa_{\mathrm{ice}}$ | 1.6 | J/(m s K) |
| thermal diffusivity ice | $\alpha_{\mathrm{ice}}$ | $0.843*10^{-6}$ | $\mathrm{m}^2/\mathrm{s}$ |
| density water and ice | $\rho$ | $10^3$ | $\mathrm{kg/m}^3$ |
| latent heat of melting | $\rho L_H$ | $0.334*10^9$ | $\mathrm{J/m}^3$ |
| Young's modulus ice | $E$ | $0.88*10^9$ | Pa |

Table 1: material constants of water and ice

The last approximation uses that $x$ is a few centimeters and $R$ about 20 meters. $v_{\mathrm{sk}}$ is also the velocity with which the top of the water layer, in contact with the skate, comes down. Later on we need also $v_{\mathrm{ice}}(x)$, being the velocity at the bottom of the layer with which the ice recedes due to the pressure.

The thickness of the water layer at a point $x$ is denoted by $h(x,y)$. So in the downward direction we have the skate between $0 < z < d(x)$, water between $d(x) < z < d(x) + h(x,y)$ and ice below $z > d(x) + h(x,y)$. The water at the sides of the skate is of minor influence, since the depth $d(x)$ measures in $\mu$m, while the width of the skate is around 1 mm. In order to focus on the essentials we restrict the discussion to the treatment of the layer underneath the skate. In Fig. 2 we give a sketch of the transverse cross-section in the $y,z$ plane. As indicated in this figure, the water layer may vary in the transverse $y$ direction. In the coming sections we approximate $h(x,y)$ by a function $h(x)$ of $x$ alone. In Appendix B, we show that this is a good approximation for calculating the friction.

## 3 Material constants of water and ice

In the Table 1 we have listed the relevant material constants of water and ice. Apart from these well known constants, there are two more material properties relevant for skating: the hardness of ice $p_H$ and the deformation rate $\gamma$. The Brinell hardness number is measured by pushing with a force $F$, an "undeformable" spherical ball into the material. After lifting the force, the material shows a dent, with surface $S$. The ratio $F/S$ is independent of $F$ and equal

to the hardness $p_H$. This means that the material reacts upon deforming forces with a fixed counter pressure $p_H$, such that the contact surface $S$ times $p_H$ balances the applied force $F$.[1]

For the hardness dependence of ice on the temperature Pourier et al. [10] give the relation

$$p_H = (14.7 - 0.6\,T) * 10^6 \,\text{Pa}, \tag{5}$$

with $T$ the temperature in centigrades. An earlier measurement gave quite different values [11]. The value depends on the method of measurement [5]. We take the viewpoint that the hardness is defined by the response to a quasi-static deformation of the ice. Mostly the hardness comes into our analysis as a multiplicative constant. Although the measurements of Pourier et al. were not carried out quasi-statically, we stick to the value given in Eq. (5) for the hardness in our calculations, when explicitly needed.

However, skating is a dynamic event. For instance a forward skating velocity of 10 m/s implies, a downward velocity of about 1 cm/s at the tip of contact. In order that the ice recedes at such a large rate, one needs pressures far exceeding the hardness. Such large pressures require a relation between the applied pressure and the velocity with which the ice recedes. With $p(x, y, d(x) + h(x))$ the pressure in the water layer in contact with the ice, we will use for the downward velocity of ice the relation

$$v_{\text{ice}}(x, y) = \gamma[p(x, y, d(x) + h(x)) - p_H], \tag{6}$$

where $\gamma$ is a material constant with the dimension $[\text{m}/(\text{Pa s})]$. Eq. (6) takes the receding velocity proportional to the pressure excess. This is similar to treating ice as a Bingham solid [12], where one puts, for plastic flow, the shear rate proportional to pressure excess. The deformed region of the ice is of the order of the width $w$. So dividing $v_{\text{ice}}$ by $w$ gives the order of the occurring shear rates. In this way we deduce, from the measured shear rates [13,14], a value $\gamma p_H \simeq 1$ mm/s. This is not more than an order of magnitude estimate, since glaciers and laboratory experiments induce plastic flows on a time scale much lower than in speed skating.

## 4 Static deformation

Elastic deformations of ice are controlled by the elastic coefficient (Youngs modulus). By calculating the elastic deformation field due to a skate which bears a weight $M$, one estimates that for $M$ below 10kg, the skate makes an elastic deformation. The estimate is hampered by role of the edges of the skate. If they are not rounded off a bit, they produce a kink in the deformation field, which leads to unlimited pressures in the ice. The estimate shows, however, that for practical skater masses the deformation is plastic.

Static inelastic deformations are determined by the hardness $p_H$. At rest, the skater exerts a pressure on the ice equal to the hardness $p_H$. The contact area times the pressure balances the weight of the skater. The contact area is the width $w$ of the skate times the contact length $2l_0$. So one has the force balance

$$Mg = 2p_H w l_0, \tag{7}$$

which gives the value of $l_0$. The static depth $d_0$ of the dent in the middle of the skate is related to $l_0$ by geometry

$$R^2 = (l_0)^2 + (R - d_0)^2, \qquad \text{or} \qquad d_0 \approx \frac{l_0^2}{2R}. \tag{8}$$

---

[1]The Brinell hardness takes as contact surface the spherical surface of the dent, which is slightly larger than the top circle of the dent. In contrast to the Brinell hardness, we measure the contact area in the horizontal direction and not along the skate, since the horizontal surface matters for the force balance Eq. (2).

The two equations (7) and (8) determine the static values of $l_0$ and $d_0$. We find for a weight of 72 kg the values $l_0 = 2.2$cm and $d_0 = 11\mu$m. We note that this estimate assumes that the pressure distribution in the ice underneath the skate is uniformly equal to $p_H$. If one calculates, for small weights, the pressure distribution for elastic deformations, one finds that the pressure is largest at the edges of the skate and in the middle where the deformation is deepest. Thus at the point where the elastic deformation turns gradually into a plastic deformation the above estimate does not apply. It only applies for a fully developed plastic deformation.

The calculation of contact length $l$ and the depth $d$ for a moving skater is a major part of the problem. The relation between $l$ and $d$ is the same as Eq. (8) between $l_0$ and $d_0$, since it is geometric. We will see that for a fast moving skater the contact length $l$ is substantially shorter than the $2l_0$ needed at rest. While for static contact the total length, forward and backward, $2l_0$ counts, for the dynamic contact only the forward section $0 \leq x \leq l$ is relevant. What happens in the backward section $-l \leq x < 0$ does not contribute to the heat balance nor to the friction, since the contact between ice and skate is broken.

## 5 The heat balance

The heat generated by friction in the water layer leads to melting of ice. The first point for establishing the heat balance is to compute the melting velocity $v_{\mathrm{m}}(x)$. The trough made by the skate has a width $w$ and a depth $d(x) + h(x)$. So the trough grows downwards at a rate

$$v_{\mathrm{tr}} = \left( \frac{\partial [d(x,t) + h(x,t)]}{\partial t} \right)_{t=0}. \tag{9}$$

Since the trough grows by melting with a velocity $v_{\mathrm{m}}$ and ploughing, which has a downward velocity $v_{\mathrm{ice}}$, we have the equality

$$v_{\mathrm{m}}(x) + v_{\mathrm{ice}}(x) = v_{\mathrm{tr}}. \tag{10}$$

Working out the right hand side of Eq. (9) gives the expression for the melting velocity

$$v_{\mathrm{m}}(x) = v_{\mathrm{sk}}(x) - v_{\mathrm{ice}}(x) - V\frac{\partial h}{\partial x}, \tag{11}$$

with $v_{\mathrm{sk}}$ given by Eq. (4) and $v_{\mathrm{ice}}$ by Eq. (6).

The main source of heat is the friction in the water layer due to the gradient in $v_x$. The gradient of the transverse flow $v_y$ contributes an order of magnitude less to the heat generation. So the frictional heat generated in a time $dt$ and a volume $h(x)w\,dx$ equals

$$dH(x) = \eta \frac{V^2}{h^2(x)} h(x) w\,dx\,dt. \tag{12}$$

The heat gives rise to melting of a volume $dV(x)$, but it is a delicate question which fraction of the heat is effective. There are two competitors for melting. Inside the water layer a fraction $\zeta_{\mathrm{w}}$ will flow towards the ice and the remainder will flow towards the skate. In Appendix D it is shown that the fraction $\zeta_{\mathrm{w}} \geq 1/2$, but usually equal to 1/2, when the difference between skate and ice temperature is small. The second competitor is the heat flow inside the ice, which is a subtle point, playing a role at low-temperature skating. We discuss this effect in Section 13. We stick here to the fraction 1/2 and get for the molten volume

$$d\mathcal{V}(x) = \frac{dH(x)}{2\rho L_H}, \tag{13}$$

with $\rho L_H$ the latent heat per volume. Equating this molten volume with the increase in water due to $v_m(x)$ leads to the balance equation

$$v_m(x)w\,dx\,dt = d\mathcal{V}(x) = k\frac{V}{h(x)}w\,dx\,dt, \tag{14}$$

where $k$ is the important parameter introduced by Le Berre and Pomeau [9]

$$k = \frac{\eta V}{2\rho L_H}. \tag{15}$$

$k$ is a (microscopic) small length. We find for skating conditions $k = 2.1 * 10^{-11}$m.[2]

We now turn this equation into a differential equation for $h(x)$ by substituting Eq. (11) into Eq. (14). Bringing the difference $v_{sk} - v_{ice}$ to the right hand side yields

$$-V\frac{\partial h}{\partial x} = k\frac{V}{h(x)} - [v_{sk}(x) - v_{ice}(x)]. \tag{16}$$

This equation becomes useful if we have an expression for the receding velocity $v_{ice}(x)$. For the ice to recede, the water layer must have a pressure $p$ exceeding the hardness $p_H$ of ice. The pressure in the water layer will be lower than $p_H$ near the midpoint $x = 0$, where the layer is close to the open air. We will show that near the tip $x = l$ the pressure will exceed $p_H$. We call the fraction with $p > p_H$ the ploughing regime and the fraction with $p < p_H$ the melting regime.

In the melting phase we have $v_{ice}(x) = 0$ and with $v_{sk}(x)$ from Eq. (4), we get the layer equation

$$-\frac{dh(x)}{dx} = \frac{k}{h(x)} - \frac{x}{R}, \tag{17}$$

which is the equation derived by Le Berre and Pomeau [9].

In the ploughing phase we need the expression Eq. (6) for the receding speed $v_{ice}$. The pressure in the water layer has to depend on $y$, since it drives out the water sideways. This causes the receding velocity to depend on $y$ and that in turn makes the layer thickness $h$ also dependent on $y$. In order to stick to the approximation where $h$ depends only on $x$, we replace Eq. (6) by its average over $y$

$$v_{ice}(x) = \gamma\int_{-w/2}^{w/2}\frac{dy}{w}[p(x, y, d(x) + h(x)) - p_H]. \tag{18}$$

In Appendix B it is outlined how the $y$ dependence in $v_{ice}$ can be accounted for.

Eq. (17) is derived without information about the hydrodynamics of the water layer, other than that the gradient in $v_x$ is the main source of friction. In the ploughing regime, where $v_{ice}(x) \neq 0$ we have to resolve the pressure dependence from the flow pattern.

## 6 The melting regime

In order to analyse the layer equation (17), we introduce two length scales as a combination of the microscopic length $k$ and the macroscopic length $R$. The longitudinal length $s_l$ and the depth length $s_d$ are defined as

$$s_l = (kR^2)^{1/3}, \qquad \text{and} \qquad s_d = (k^2R)^{1/3}. \tag{19}$$

---

[2]Actually $k$ is about a factor $10^3$ smaller than the value $1.8*10^{-8}$m given by the authors of [9], since they erroneously take for the water density $\rho = 1$, while $\rho = 10^3$ in SI-units.

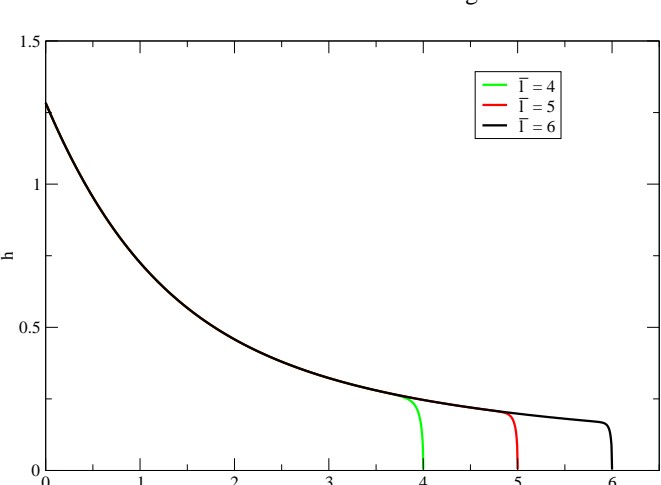

Figure 3: The scaled thickness $\bar{h}$ of the layer as a function of the scaled position $\bar{x}$ in the melting regime. The curves are evaluated for some values of the scaled contact length $\bar{l}$. For negative $\bar{x}$ the water layer is irrelevant. It may be given the constant value $\bar{h}(\bar{x}) = \bar{h}(0)$.

For the skating velocity $V = 8$m/s, we have as the scale for the contact length $s_l = 2.16$ mm and as scale for the thickness $s_d = 0.21\mu$m. Both are rather small.[3]

If we use $s_l$ as a scale for the longitudinal coordinate $x$ and $s_d$ for the thickness $h$

$$x = s_l \bar{x} \qquad \text{and} \qquad h = s_d \bar{h}, \tag{20}$$

Eq. (17) becomes dimensionless

$$-\frac{d\bar{h}(\bar{x})}{d\bar{x}} = \frac{1}{\bar{h}(\bar{x})} - \bar{x}. \tag{21}$$

The advantage of this scaled equation is that no external parameters occur in the equation. The skating velocity $V$ and radius of curvature $R$ come in via the scales $s_l$ and $s_d$ through the parameter $k$.

Eq. (21) is easy to integrate numerically, starting from a guess for the contact length $\bar{l}$. At $\bar{x} = \bar{l}$ the thickness $\bar{h}$ vanishes and thus the first term on the right hand side of Eq. (21) dominates and the solution behaves as

$$\bar{h}(\bar{x}) \simeq \sqrt{2(\bar{l} - \bar{x})}, \qquad \bar{x} \to \bar{l}. \tag{22}$$

In Fig. 3 we have given the curves for a few values of $\bar{l}$. The curves distinguish themselves only near the tip $\bar{x} = \bar{l}$. Integrating the equation from below starting at $\bar{x} = 0$, there is a value $\bar{h}_0 \simeq 1.284$ such that the curves with $\bar{h}(0) > \bar{h}_0$ curve upwards asymptotically and the curves with $\bar{h}(0) < \bar{h}_0$ bend downwards hitting the axis. The seperatrix starting at $\bar{h}(0) = \bar{h}_0$ behaves asymptotically as $\bar{h}(\bar{x}) \simeq 1/\bar{x}$.

The value of the contact length follows from the balance between the pressure in the water layer and the weight $M$ of the skater, for which we need the pressure in the water layer.

---

[3]The water layer thickness $s_h$ would multiply with a factor 100 for $\rho = 1$ and the length $s_l$ with a factor 10. These values are comparable with the values found by Le Berre and Pomeau [9].

## 7 The hydrodynamics of the water layer

The pressure is determined by the hydrodynamic equations of the water layer. The pressure distribution has been derived both in [8] and [9]. Here we give the expressions which are important for the next section. In Appendix A we sketch how the pressure follows from the assumption that the transverse flow has a Poisseuille form

$$v_y(x, y, z) = a(x) y [z - d(x)][h(x) - z + d(x)].$$  (23)

The amplitude $a(x)$ determines, through the fluid equations, the pressure behaviour. At the top and bottom of the layer we have

$$p(x, y, d(x)) = p(x, y, d(x) + h(x)) = \eta a(x) \left( \frac{w^2}{4} - y^2 \right).$$  (24)

The pressure is maximal in the middle of the skate blade and drops off towards the edges. The $y$ dependence of the pressure is essential for pushing out the water towards the edges of the skate. (It causes also an $y$ dependence in the layer thickness $h$, see Appendix B.)

The incompressibility of water implies the connection of $a(x)$ with the downward velocities of the top and bottom of the water layer

$$v_{\text{sk}}(x) - v_{\text{ice}}(x) = a(x)h^3(x)/6$$  (25)

Eq. (25) holds both in the melting and the ploughing phase. In the melting regime, where $v_{\text{ice}} = 0$, it implies a simple relation between $a(x)$ and $h(x)$

$$V \frac{x}{R} = a(x)h^3(x)/6.$$  (26)

Using Eq. (26) for $a(x)$ and Eq. (24), gives for the average pressure in the melting phase the expression

$$\frac{1}{w} \int_{-w/2}^{w/2} dy \, p(x, y, d(x) + h(x)) = \frac{\eta w^2 V}{R} \frac{x}{h^3(x)}.$$  (27)

This presents a problem for the weight balance, if the melting phase would apply all the way to the tip, where $h(x)$ behaves as given by Eq. (22). That leads to a diverging pressure, which is non-integrable. So some regularisation near the tip is necessary, see [9]. In our treatment this problem does not occur, since the the ploughing regime takes over as soon as the pressure exceeds the hardness $p_H$.

## 8 The ploughing regime

As follows from the analysis of the previous section, part of the deformation of ice is due to the force on the ice. With Eq. (24) and Eq. (18) we find

$$v_{\text{ice}}(x) = \gamma [\eta a(x) w^2/6 - p_H].$$  (28)

Using this expression in Eq. (25) we obtain the following relation between $a(x)$ and $h(x)$

$$a(x) = 6 \frac{V x/R + \gamma p_H}{h^3(x) + \gamma \eta w^2}.$$  (29)

The heat balance equation (16) can be cast, with Eq. (25), into the form

$$-\frac{dh(x)}{dx} = \frac{k}{h(x)} - \frac{a(x)h^3(x)}{6V}.$$  (30)

Then using $a(x)$ from Eq. (29), turns it into an explicit layer equation for $h(x)$

$$-\frac{dh(x)}{dx} = \frac{k}{h(x)} - \frac{x/R + \gamma p_H/V}{h^3(x) + \gamma \eta w^2} h^3(x). \tag{31}$$

We note that putting $\gamma = 0$, which is equivalent to putting $v_{\text{ice}} = 0$, reduces indeed the equation to Eq. (17) of the melting regime. On the other hand, the limit $\gamma \to \infty$ reduces the equation to

$$-\frac{dh(x)}{dx} = \frac{k}{h(x)} - \frac{p_H}{\eta w^2 V} h^3(x), \tag{32}$$

which is the backbone of the equation derived by Lozowski and Szilder [8]. A very large $\gamma$ implies that the pressure at the bottom of the layer stays equal to the hardness $p_H$ and that is an implicit assumption in [8]. Eq. (32) can be solved analytically, see Appendix C.

In order to get a better insight in Eq. (31), we make the equation dimensionless by introducing the same scaling as in Eq. (20), yielding the layer equation

$$-\frac{d\bar{h}}{d\bar{x}} = \frac{1}{\bar{h}(\bar{x})} - \frac{\bar{x} + c_1}{c_2 + \bar{h}^3(\bar{x})} \bar{h}^3(\bar{x}), \tag{33}$$

with the dimensionless constants

$$c_1 = \frac{\gamma p_H}{V} \left(\frac{R}{k}\right)^{1/3}, \qquad c_2 = \frac{\gamma \eta w^2}{k^2 R}. \tag{34}$$

The magnitude of these constants depends on the value of $\gamma$, on which we have little experimental evidence. With the value $\gamma p_H = 10^{-3}$ m/s, we get for skating conditions

$$c_1 = 1.27, \qquad c_2 = 15.0 \qquad c_3 = \frac{c_1}{c_2} = 0.085. \tag{35}$$

Note that the ratio $c_3$ is independent of $\gamma$.

## 9 The cross-over from ploughing to melting

We must integrate Eq. (33) starting from a value $\bar{l}$ till a point where the velocity $v_{\text{ice}}(x)$ tends to become negative. Thus with Eq. (28) we have to obey the condition

$$\eta a(x) w^2 > 6 p_H. \tag{36}$$

With the expression (29) for $a(x)$ this translates to

$$\eta w^2 V x/R > p_H h^3(x), \qquad \text{or} \qquad \bar{x} > c_3 \bar{h}^3(\bar{x}). \tag{37}$$

At the top $\bar{x} = \bar{l}$ we have $\bar{h}(\bar{l}) = 0$. So there the inequality is certainly fulfilled. At the midpoint $\bar{x} = 0$, so there the inequality is certainly violated. Somewhere in between, at the cross-over point $\bar{l}_c$, the ploughing regime merges smoothly into the melting regime. In dimensionless units, $\bar{l}_c$ is the solution of the equation

$$\bar{l}_c = c_3 \bar{h}^3(\bar{l}_c). \tag{38}$$

At the cross-over point the layer thickness $\bar{h}_c = \bar{h}(\bar{l}_c)$ is the same in both regimes. The derivative is also continuous at the cross-over point. We find in the ploughing regime

$$-\left(\frac{d\bar{h}}{d\bar{x}}\right)_{\bar{l}_c} = \frac{1}{\bar{h}(\bar{l}_c)} - \frac{c_3 \bar{h}^3(\bar{l}_c) + c_1}{c_2 + \bar{h}^3(\bar{l}_c)} \bar{h}^3(\bar{l}_c) = \frac{1}{\bar{h}(\bar{l}_c)} - c_3 \bar{h}^3(\bar{l}_c) = \frac{1}{\bar{h}(\bar{l}_c)} - \bar{l}_c, \tag{39}$$

which equals the value in the melting regime.

## 10 Scaling the pressure in the water layer

The pressure at the top of the water layer is given by Eq. (24) and with Eq. (29) for the amplitude $a(x)$ we get in the ploughing regime

$$p(x) = \eta w^2 \frac{Vx/R + \gamma p_H}{\gamma \eta w^2 + h^3(x)}. \tag{40}$$

It is interesting to compare this value with the hardness $p_H$ of ice and to express this ratio in dimensionless units

$$\bar{p}(\bar{x}) = \frac{p(x)}{p_H} = \frac{\eta w^2}{p_H/V} \frac{x/R + \gamma p_H/V}{\gamma \eta w^2 + h^3(x)} = \frac{\bar{x} + c_1}{c_1 + c_3 \bar{h}^3(\bar{x})}. \tag{41}$$

This expression holds in the ploughing regime. In the melting regime we have

$$\bar{p}(\bar{x}) = \frac{1}{c_3} \frac{\bar{x}}{\bar{h}^3(\bar{x})}. \tag{42}$$

Note that, with Eq. (38), both expressions (41) and (42) yield $\bar{p}(\bar{l}_c) = 1$. $\bar{p}(\bar{x})$ is larger than 1 in the ploughing phase and smaller than 1 in the melting phase. The maximum pressure occurs at the tip, $\bar{x} = \bar{l}$, where $\bar{h} = 0$, with the value

$$\bar{p}_t = \bar{p}(\bar{l}) = 1 + \frac{\bar{l}}{c_1}. \tag{43}$$

As $\bar{l}$ will turn out to be around 6, this is a substantial ratio.

## 11 The macroscopic forces

The skate feels a normal and tangential force. The normal force $F_N = Mg$ is the weight of the skater. The tangential friction force has two ingredients: the friction force $F_{\text{fr}}$, due to the water layer and the ploughing force $F_{\text{pl}}$, which pushes down the ice. All three forces are related to integrals over the contact zone. The weight $M$ of the skater is balanced by the pressure at the top in the water layer

$$F_N = w \int_0^l dx\, p(x). \tag{44}$$

The friction force is given by the gradient of the flow in the water layer

$$F_{\text{fr}} = \eta w \int_0^l dx \frac{V}{h(x)}. \tag{45}$$

The ploughing force results from the force that the pressure in the water layer exerts on the ice in the forward direction. It is given by

$$F_{\text{pl}} = w \int_0^l dx \frac{x}{R} p(x). \tag{46}$$

The ratio $x/R$ gives the component of the force in the forward direction.

Applying the scaling Eq. (20) on $x$ and $h(x)$ and scaling the pressure with the hardness $p_H$, we get the expressions

$$
\begin{cases}
F_N & = & a_N \displaystyle\int_0^{\bar{l}} d\bar{x}\, \bar{p}(\bar{x}), \\[2ex]
F_{\text{pl}} & = & a_{\text{pl}} \displaystyle\int_0^{\bar{l}} d\bar{x}\, \bar{x}\, \bar{p}(\bar{x}), \\[2ex]
F_{\text{fr}} & = & a_{\text{fr}} \displaystyle\int_0^{\bar{l}} d\bar{x}\, \frac{1}{\bar{h}(\bar{x})}.
\end{cases}
\tag{47}
$$

The integrals are dimensionless and the constants have the dimension of a force

$$
a_N = p_H\, w\, s_l, \qquad a_{\text{pl}} = p_H\, w\, s_d, \qquad a_{\text{fr}} = \eta\, V\, w\, \frac{s_l}{s_d}.
\tag{48}
$$

Note that the ratio $a_{\text{pl}}/a_N$ involves the ratio of the scales $s_d/s_l$, which is a reflection of the fact that the normal force acts over the longitudinal length $l$ and the ploughing over the depth $d$. In order to compare friction with ploughing, we use the number $\lambda$ introduced in Eq. (82), leading to

$$
\eta V = 2k\, \rho\, L_H = 2k\, \lambda\, p_H.
\tag{49}
$$

This gives for the relation between $a_{\text{fr}}$ and $a_{\text{pl}}$

$$
a_{\text{fr}} = 2k\, \lambda\, p_H\, w\, \frac{s_l}{s_d} = 2p_H w\, \lambda s_d = 2\lambda\, a_{\text{pl}}.
\tag{50}
$$

An interesting feature of pressures $p(x)$, exceeding the hardness $p_H$ in the ploughing regime, is that they shrink the contact length $l$ and the penetration depth $d$, since $d$ goes with the square of $l$. So the skater "rises" due to his velocity. We find in the limit $V \to 0$ an indentation depth $d \simeq 44\mu$m and for $V = 8$m/s a value $d = 4.5\mu$m.[4] For slow velocities the $F_{\text{fr}}$ vanishes and $F_{\text{pl}}$ has a limit $\simeq 0.7$ N for a skater of 72 kg. For the $V = 8$m/s we find $F_{\text{fr}} = 0.84$ N and $F_{\text{pl}} = 0.29$ N. So the large pressure build-up near the tip, reduces the ploughing force, from dominant at $V \to 0$, to a fraction of the total friction force.

## 12  Velocity dependence of the friction

The integration of the layer equation is straightforward once we know the contact length $l$. The value of $l$ determines the weight of the skater. Since the weight is given, we must find the contact length by trial and error. In Fig. 4 we have drawn the shape of the water layer for a few values of the deformation rate $\gamma$ and a skater weight of 72 kg. The curves end at $x = l$ and one observes that the contact length is rather sensitively dependent on the value of $\gamma$. This is not surprising since $\gamma$ has a direct influence on the pressure in the water layer and the pressure determines the weight. The small up-swing of the thickness in the middle of the skate ($x = 0$) is a manifestation of the melting phase. On the other hand the overal thickness of the layer does not depend strongly on the value of $\gamma$.

The next result is the friction as function of the velocity. In Fig. 5 we have drawn how the ploughing and water friction combine to the total strength of the friction. While both components vary substantially with the velocity, the combination is remarkably constant over a wide range of velocities. One observes that the low $V$ limit (exhibiting a square root dependence on $V$), covers only a very small region of velocities. In the Fig. 5 we have also plotted the

---

[4]See Appendix C for the relation between the static $d_0$ and $d$ in the slow limit.

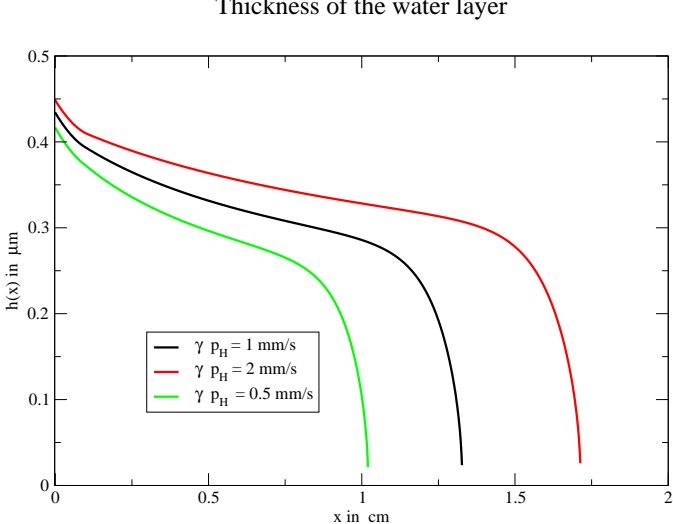

Figure 4: The shape of the water layer for some values of $\gamma$ and skating conditions

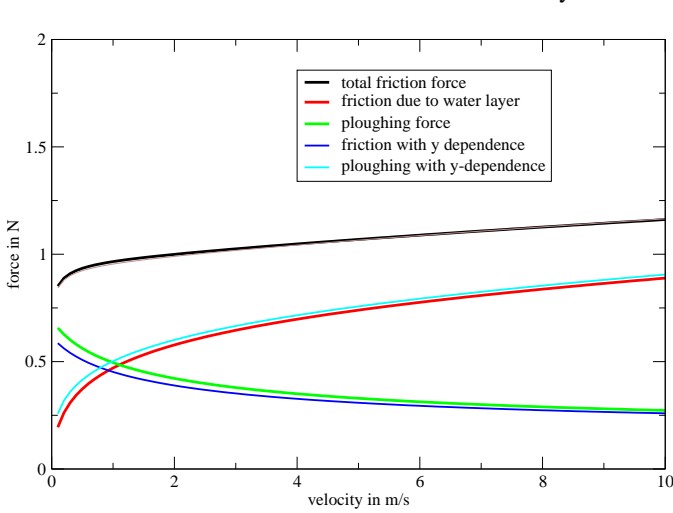

Figure 5: The various contribution to the friction as function of the velocity $V$, for otherwise skating conditions.

influence on the contributions, if one takes the $y$ dependence of the thickness into account. The effects on friction and ploughing are small and opposite, such that the change of the total friction is not visible in the Fig. 5.

In order to see how much the value of $\gamma$ influences the friction, we have drawn in Fig. 6 the total friction as function of the velocity for some values of $\gamma$. The influence of $\gamma$ is noticeable, but not dramatic. A factor 16 difference in $\gamma p_H$, between $\gamma p_H = 4$ mm/s and $\gamma = 0.25$ mm/s, gives a factor 2 in the friction for large velocities. But there is a substantial difference with respect to the theory of Lozowski and Szilder [8], using $\gamma = \infty$.

Usually the friction is expressed in terms of the friction coefficient $\sigma$, being the ratio of the

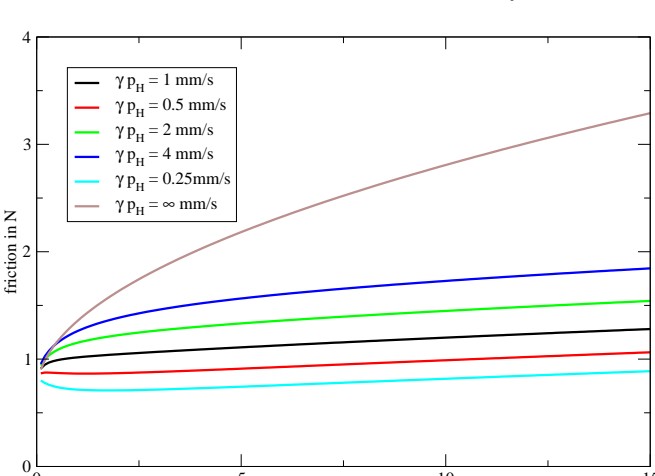

Figure 6: The total friction as function of the velocity for some values of $\gamma$

tangential and the normal force. In the present case it reads

$$\sigma = \frac{F_{\text{fr}} + F_{\text{pl}}}{F_N}. \tag{51}$$

However, for skating the friction coefficient is not independent of the normal force. In a standard friction experiment the contact surface is proportional to the normal force and the friction force is proportional to the contact area, such that in the friction coefficient the contact area drops out. This proportionality does not hold for skating. The order of magnitude of the friction coefficients is 0.002 for skating conditions. We estimate the contact area for skating conditions as $s_l \bar{l} w \simeq 14.3 \text{ mm}^2$.

## 13 Temperature dependence of the friction

So far we have considered temperatures close to the melting point of ice, where temperature gradients and associated heat flows are small. At lower temperatures they start to play a role. In order to melt ice, one first has to heat it to the melting temperature $T_m$. If the difference between the melting temperature and the surface temperature $T_s$ is positive, i.e. when the surface temperature is lower than the melting temperature, one has to increase the latent heat $L_H$ with the amount needed to heat the ice. Since the latent heat is 80 times the heat necessary to raise the temperature of ice by one degree, this is usually a small correction.

Another small correction comes from the heat flux which may exist in the ice layer. In a skating rink the ice is cooled from below and there is a heat flow downwards. Natural ice freezes by cooling the top layer and correspondingly the heat flow is upwards. But the temperature gradients are small with respect to the temperature gradients in the water layer, so the effect on the amount of ice that melts is small and we leave it out.

However, as pointed out in [8], there is another heat flow, which can have an important effect on the friction at low ice temperatures. If the surface temperature is low, one has to heat the surface, before it melts. This causes a temperature gradient in the ice and an associated heat leak into the ice. The melting occurs under pressure and one has to raise the temperature,

not to zero centigrade, but to the melting temperature $T_m$ at that pressure. Since the pressure $p$ in the water layer is large, $T_m$ can be substantial below zero degree centigrade. The lowering of the melting temperature is approximately given by

$$T_m = -0.1 p * 10^{-6}. \tag{52}$$

The maximum value of $T_m$ occurs at $p = 2 * 10^8$ Pa, producing a $T_m$ of -20 degrees centigrade. Since the pressure varies strongly with the position $x$ of the contact, $T_m$ varies also with $x$. In the middle of the skate, where the contact ends, the pressure vanishes and the melting temperature $T_m(0) = 0$. At the tip the pressure is maximal and $T_m(l)$ reaches its lowest point. We have to distinguish two cases: $T_m(l) < T_s$ and $T_m(l) > T_s$. In the former case, there is a point $x_0$ where $T_m(x_0) = T_s$. For $x > x_0$ the melting temperature is then below the surface temperature and no heat is needed to raise the ice to $T_m$. In the latter case the ice is heated all along the contact line and at the tip a sudden jump in the surface temperature occurs.

In Appendix E we have given the derivation of the temperature gradient in the ice at the surface. It reads

$$\left(\frac{\partial T(x,z)}{\partial z}\right)_{z=0} = \left(\frac{V}{\pi \alpha_{\text{ice}}}\right)^{1/2} \left(-\int_x^{x_0} dx' \frac{1}{\sqrt{x'-x}} \frac{\partial T_m(x')}{\partial x'} + \frac{T_m(l) - T_s}{\sqrt{l-x}}\right). \tag{53}$$

The understanding is that the last term is absent for $T_m(l) < T_s$ and in the other case the integral extends to $x_0 = l$. In [8] only this last term is taken into account, together with setting $T_m(l) = 0$.

The gradient gives a downward heat flow at the surface $z = 0$

$$J_{\text{ice}}(x) = -\kappa_{\text{ice}} \left(\frac{\partial T(x,z)}{\partial z}\right)_{z=0}. \tag{54}$$

This gradient takes away a fraction of the heat supplied by $J_{\text{w-ice}}$

$$\zeta(x) = 1 - \frac{J_{\text{ice}}(x)}{J_{\text{w-ice}}}, \tag{55}$$

with $J_{\text{w-ice}}$ given by Eq. (95).

In the layer equation we have to replace the first term of the right hand side by

$$\frac{k}{h(x)} \rightarrow \frac{k\zeta(x)}{h(x)}. \tag{56}$$

In Fig. 7 we have drawn the friction as function of the surface temperature for skating conditions. Note that the friction hardly changes in the region $0 > T_s > -5^0$C, after which the friction starts to increase. The influence of $\gamma$ is similar for all temperatures.

# 14 Discussion

We have investigated the thickness of the water layer underneath the skate as a result of melting of ice by the frictional heat. In skating two processes take place: a plastic deformation of the ice (ploughing) and the generation of a water layer by melting. For low velocities ploughing dominates and for high velocities friction in the water layer dominates. In the skating range of velocities, the total friction is rather independent on the velocity. The friction in the water layer increases with the velocity, which is compensated by an almost equal decrease in the ploughing force. A high skating velocity causes a high pressure in the water layer, lifting the skater. Consequently the skate penetrates less deep into the ice and the ploughing force

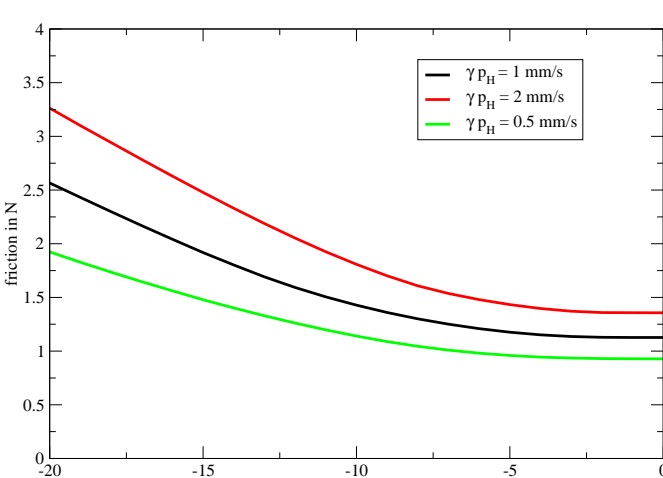

Figure 7: The friction for skating conditions as function of the temperature.

decreases. The theory assumes that the thickness of the water layer is large enough to treat it hydrodynamic system. For low velocities and low temperatures this assumption breaks down (see Appendix E).

There are two important material constants of ice, determining the friction: the hardness $p_H$ and the deformation rate $\gamma$. Unfortunately no accurate data exist on these constants, which hampers a quantitative calculation of the friction. In particular the value of $\gamma$ is poorly known, while it has a substantial influence on the magnitude of the friction. The relevance of $\gamma$ becomes clear from an estimate of the speed at which ice has to be pushed down at the tip of contact. For a forward velocity of 10 m/s, the downward speed of the ice is about 1 cm/s. Such high deformation rates require large pressures, several times the hardness.

The most important theoretical parameter is $k$, defined in Eq. (15), which is microscopically small for reasonable values of the velocity of the skate. In combination with the macroscopic curvature $R$ of the skate, two length scales follow: the longitudinal scale $s_l$ and the depth scale $s_d$ defined in Eq. (19). $s_l$ is a measure for the contact length and $s_d$ gives the magnitude of the thickness of the water layer.

Our analysis combines elements of the theory of Le Berre and Pomeau [9], which only accounts for the effects of melting and the theory of Lozowski and Szilder, which equals the pressure in the water layer to the hardness of ice. The new element is that we propose that the ice recedes with a velocity proportional to the excess pressure with respect to the hardness. In the theory of Le Berre and Pomeau the ice does not recede (which is equivalent with $\gamma = 0$), in spite of the fact that, in their approach, the pressure grows unlimitedly near the tip. In the theory of Lozowski and Szilder the ice adapts instantaneously (which is equivalent with $\gamma = \infty$), keeping the pressure equal to the hardness.

We have mainly considered skating near the melting point. At low temperatures a number of new elements come into play, which we have indicated in Section 13. A quantitative discussion of these effects is delicate, since they depend not only on the conditions of the ice, but also on the value of the constant $\gamma$ in the Bingham Eq. (6). Since the hardness $p_H$ and the deformation rate $\gamma$, are not very well known as function of the temperature, a precise measurement of these properties would be very welcome.

We have left out a number of refinements in order to focus on the essential features of

skating. Refinements that can be treated in the presented context are:

- We have omitted the influence of the melting of the ice at the sides of the skate. A simple treatment adds to the width $w$ on both sides the amount $d(x)$. Since the indentation depth $d$ of the skate is very small compared to the width $w$ of the blade (we find a ratio $1/500$) it gives a small correction.

- We have assumed that only the gradient of the forward velocity contributes to the friction and the corresponding heat generation. It is easy to take into account the contributions of the gradient in the transverse velocity. The relative importance of the longitudinal and transverse heat generation is of the order $1/\lambda$, see Eq. (82). This means that the transverse velocity gradient contributes only a few percent to the generated heat.

- Most of our calculations are based on the assumption that the thickness $h$ depends only on the longitudinal coordinate $x$. In Appendix B we have made a start of taking the transverse $y$ dependence into account. A fully consistent treatment, including the hydrodynamic equations, is computationally quite involved and as far as the friction is concerned not very encouraging, as the effect is quite small in lowest order (see Fig. 5). The reason is that the variation of $h(x, y)$ with $y$ is modest except at the edges of the skate.

There are several influences outside our scope, such as the humidity of the air and the addition of suitable chemicals to the surface layer, which are important for speed skating records, but not essential for the phenomenon of skating. Apart from a more accurate measurement of $p_H$ and $\gamma$, it would be interesting if the deformation of the ice, due to the skate, could be observed. Presumably the 10% difference in density between ice and water, which we ignored, plays an important role for the form of the deformation.

De Koning et al. [2] report a friction force of 3.8 N for the straight strokes and 4.9 N for the curves. The difference is due to the fact that in the curves the skate is at an angle with the ice. In the straights there are also parts, at the begin and end of the stroke, where the skate makes an angle with the ice. So for the upright part, for which we perform the calculation, one estimates a friction force around 2 N. This compares well with the values we see in Fig. 6. A fit might be seen as a measurement of $\gamma$ and tends to the value $\gamma p_H = 2$ mm/s.

In Appendix C we discuss the slow velocity limit $V \rightarrow 0$, which is hardly relevant for skating, but may be useful for measurements in the laboratory, involving low $V$.

## Acknowledgments

The author is indebted to Tjerk Oosterkamp for drawing his attention to the problem and for careful reading of the manuscript and to Tom van de Reep for explaining the details of the measurements in Leiden. He also acknowledges discussions with the experimental group of Daniel Bonn in Amsterdam, in particular the discussions with Bart Weber on ploughing. Numerous conversations about the properties of ice with Henk Blöte are highly appreciated.

## A   Velocity and pressure in the water layer

In this Appendix we discuss the hydrodynamics in the water layer underneath the skate. We take advantage of the fact that we have three different length scales: in the $x$ direction the scale is in centimeters, in the $y$ direction in millimeters and in the $z$ direction in microns. So

the gradients in the $z$ direction are much larger than in the other directions and we may use the lubrication approximation of the Navier-Stokes equations for an incompressible fluid

$$\nabla p = \eta \, \Delta \mathbf{v}, \qquad \text{and} \qquad \nabla \cdot \mathbf{v} = 0. \tag{57}$$

The velocity in the $x$-direction is forced by the motion of the skate

$$v_x = V \left( 1 - \frac{z - d(x)}{h(x)} \right). \tag{58}$$

At the top of the layer $z = d(x)$, the velocity of water equals that of the skate and at the bottom, $z = d(x) + h(x)$, it vanishes at the solid ice surface.

The velocity in the $y$ direction has a Poisseuille form

$$v_y(x, y, z) = a(x) \, y \, [z - d(x)][h(x) - z + d(x)], \tag{59}$$

This velocity component vanishes at the skate blade $z = d(x)$ as well as at the bottom of the layer at $z = d(x) + h(x)$. The linear dependence on $y$ is a consequence of the incompressibility of water. To see this, consider a volume between $x$ and $x + \delta x$, $y$ and $y + \delta y$ and $z = d(x)$ and $z = d(x) + h(x)$. At the top it goes down with the velocity $v_{\mathrm{sk}}(x)$ and at the bottom it may go down with a velocity $v_{\mathrm{ice}}(x)$. The total decrease of the volume due to vertical motion of the top and bottom boundary equals

$$\Delta V_v = [v_{\mathrm{sk}}(x) - v_{\mathrm{ice}}(x)] \delta x \, \delta y \, \delta t. \tag{60}$$

In the horizontal direction we have an inflow at $y$ and an outflow at $y + \delta y$ resulting in the net displaced volume

$$\int_{d(x)}^{h(x)+d(x)} dz [v_y(x, y + \delta y, z) - v_y(x, y, z)] \delta x \delta t = a(x) h^3(x) \delta y \delta x \delta t / 6. \tag{61}$$

As water is incompressible we have the balance

$$v_{\mathrm{sk}}(x) - v_{\mathrm{ice}}(x) = a(x) h^3(x) / 6. \tag{62}$$

The linear dependence of $v_y$ on $y$ makes the right hand side in Eq. (61) independent of $y$.

The third component of the velocity is given by

$$v_z = v_{\mathrm{ice}}(x) + a(x) \left( \frac{[z - d(x)]^3}{3} - \frac{[z - d(x)]^2 h(x)}{2} + \frac{h^3(x)}{6} \right). \tag{63}$$

Note that we have chosen the constants such that at the top $v_z(x, y, d(x)) = v_{\mathrm{sk}}(x)$ and at the bottom $v_z(x, y, d(x) + h(x)) = v_{\mathrm{ice}}(x)$.

The pressure distribution compatible with this flow field is fixed up to a constant. Here we take the boundary condition

$$p(x, w/2, d(x)) = 0, \tag{64}$$

using that at the corners of the furrow the pressure is (nearly) zero. This gives the pressure the form

$$p(x, y, z) = \eta \, a(x) \left( \frac{w^2}{4} - y^2 - [z - d(x)][d(x) + h(x) - z] \right). \tag{65}$$

At the top and the bottom the pressure equals

$$p(x, y, d(x)) = p(x, y, d(x) + h(x)) = \eta a(x) \left( \frac{w^2}{4} - y^2 \right). \tag{66}$$

which is maximal in the middle of the skate blade.

It is easy to verify that the flow field and the pressure fulfil the Navier-Stokes equations (57), provided that we consider for the differentiation only the explicit $y$ and $z$ dependence and ignore the $x$ dependencies of $a(x)$ and $h(x)$ for the calculation of the gradients.

## B The $y$ dependence of the water layer

We see from Eq. (65) that the pressure depends explicitly on $y$. This implies, through the expression (6) for $v_{\text{ice}}$, that also $v_{\text{ice}}$ is dependent on $x$ and $y$. That in turn forces the function $a$ and $h$ to depend also on $x$ and $y$. Taking the $y$ dependence fully into account, also for the detailed solution of the hydrodynamic equations in the layer of varying thickness, is quite involved. Here we give a first step, which focusses on the explicit $y$ dependence that enters into the equations. With Eq. (6) and (65) one has

$$v_{\text{ice}}(x,y) = \gamma[\eta a(x,y)(w^2/4 - y^2) - p_H]. \tag{67}$$

If $v_{\text{ice}}$ depends on $x$ and $y$, we also must change the layer Eq. (32) into

$$-\frac{\partial h(x,y)}{\partial x} = \frac{k}{h(x,y)} - \frac{1}{V}[v_{\text{sk}}(x) - v_{\text{ice}}(x,y)]. \tag{68}$$

Finally the connection between $a$ and $v_{\text{ice}}$, as given by Eq. (61), changes into

$$v_{\text{sk}}(x) - v_{\text{ice}}(x,y) = \int_{d(x)}^{h(x,y)+d(x)} dz \frac{\partial v_y}{\partial y} = \frac{1}{6}\frac{\partial}{\partial y}a(x,y)yh^3(x,y). \tag{69}$$

The last step uses that $v_y$ vanishes at the boundaries. The three equations (67)-(69) determine the behaviour of the three quantities $a(x,y), h(x,y)$ and $v_{\text{ice}}(x,y)$. We first eliminate $v_{\text{ice}}$ by inserting Eq. (67) into Eqs. (68) and (69), which leads to the set

$$\begin{cases} -\dfrac{\partial h(x,y)}{\partial x} &= \dfrac{k}{h(x,y)} - \left(\dfrac{x}{R} + \dfrac{\gamma p_H}{V}\right) + \dfrac{\gamma\eta}{V}a(x,y)[w^2/4 - y^2], \\[2mm] \dfrac{x}{R} + \dfrac{\gamma p_H}{V} &= \dfrac{1}{6V}\dfrac{\partial}{\partial y}a(x,y)y\,h^3(x,y) + \dfrac{\gamma\eta}{V}a(x,y)[w^2/4 - y^2]. \end{cases} \tag{70}$$

The equations simplify in the center $y = 0$ where we may use

$$\frac{\partial}{\partial y}a(x,y)y\,h^3(x,y) \simeq a(x,y)h^3(x,y). \tag{71}$$

In the anticipation that the variation of $a$ and $h$ with $y$ is modest, we use the approximation (71) for the whole width. Then $a(x,y)$ can be expressed in terms of $h(x,y)$ as

$$a(x,y) = \frac{6(Vx/R + \gamma p_H)}{h^3(x,y) + \gamma\eta[3w^2/2 - 6y^2]} \tag{72}$$

For calculational purpose we give the scaled version of the equations, using for $y$ and $a$ the scaling

$$y = w\bar{y}, \qquad a(x,y) = \frac{V}{k^2 R}\left(\frac{k}{R}\right)^{1/3}\bar{a}(\bar{x},\bar{y}). \tag{73}$$

For $\bar{a}$ expression (72) becomes

$$\bar{a}(\bar{x},\bar{y}) = \frac{6(\bar{x} + c_1)}{\bar{h}^3(\bar{x},\bar{y}) + c_2(3/2 - 6\bar{y}^2)}. \tag{74}$$

The constants $c_1$ and $c_2$ are defined in Eq. (34). With this value of $a$ inserted into the first Eq. (70), we get for $\bar{h}$ the equation

$$-\frac{\partial \bar{h}(\bar{x},\bar{y})}{\partial \bar{x}} = \frac{1}{\bar{h}(\bar{x},\bar{y})} - \frac{(\bar{x} + c_1)\bar{h}^3(\bar{x},\bar{y})}{\bar{h}^3(\bar{x},\bar{y}) + c_2(3/2 - 6\bar{y}^2)}. \tag{75}$$

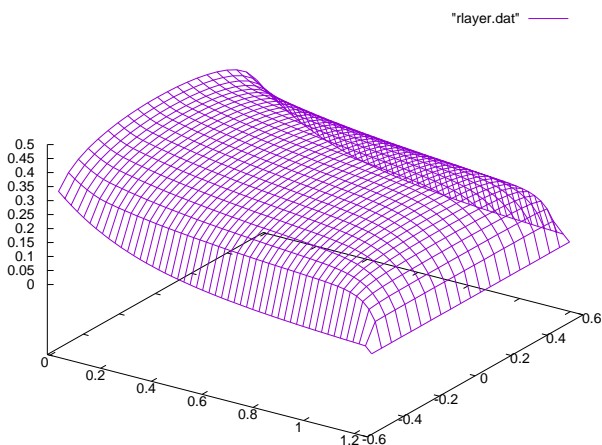

Figure 8: The vertical axis gives the thickness $h(x, y)$ of the water layer in $\mu$m. In the horizontal direction the (longer) $x$ coordinate is measured in cm and the $y$ coordinated in mm.

This equation has to be solved, starting from a value $\bar{x} = \bar{l}$, where the thickness behaves as indicated in Eq. (22). The transition to the melting regime occurs at

$$\bar{x}(3/2 - 6\bar{y}^2) = c_3 \bar{h}^3(\bar{x}, \bar{y}). \tag{76}$$

From there on the equation reads in the melting regime as before

$$-\frac{\partial \bar{h}(\bar{x}, \bar{y})}{\partial \bar{x}} = \frac{1}{\bar{h}(\bar{x}, \bar{y})} - \bar{x}. \tag{77}$$

In contrast to the equation where the average pressure was employed, the transition from the ploughing to the melting regime is $y$ dependent. It occurs immediately at the edges $\bar{y} = \pm 1/2$ and lastly in the middle $\bar{y} = 0$. In Fig. 8 we have plotted the solution of Eqs. (75) and (77) for skating conditions. Only at the edges there is a substantial $\bar{y}$ dependence. It is a consequence of the boundary condition that the pressure should vanish at the edges of the skate. In fact the pressure is always higher than the atmospheric pressure, but that is a small value as compared to the hardness of ice, which is the scale for the pressure in the water layer.

The approximation Eq. (71) can be improved by computed the derivatives of $\bar{a}$ and $\bar{h}$ from the solution of Eqs. (75) and (77) and adding that as a correction to Eq. (71). In view of the small influence on the friction by the first approximation outlined in this section, (see Fig. 6), such a further refinement is not worth while.

## C  The slow velocity limit

In this Section we discuss the limit of the velocity $V \to 0$. When the velocity $V$ of the skate becomes small, the scaling used in the previous sections is not adequate because the scales $s_l$ and $s_d$ vanish in the limit of $V \to 0$. The constants $c_1$ and $c_2$, on the other hand start to diverge as

$$c_1 \sim V^{-4/3}, \qquad \text{and} \qquad c_2 \sim V^{-2}. \tag{78}$$

Using these limits in Eq. (41) for $\bar{p}$ we see that $\bar{p}$ approaches 1, implying that for low velocities the pressure in the water layer hardly rises above the hardness $p_H$. Consequently the ice will recede also slowly. But if the pressure equals $p_H$, Eq. (32) of Lozowski and Szilder [8] becomes valid.

Fortunately Eq. (32) can be solved exactly. Using that the water layer vanishes at the top $x = l$ of the skate yields the expression for the layer

$$h(x) = A[\tanh((l-x)/l_a)]^{1/2}. \tag{79}$$

The asymptotic thickness $A$ of the layer is given by

$$A = \left(\frac{\eta^2 w^2}{p_H \rho L_H}\right)^{1/4} V^{1/2} = (k^2 \lambda)^{1/4} \tag{80}$$

and the length $l_a$ of the onset of the asymptotic value reads

$$l_a = \frac{w}{2}\left(\frac{\rho L_H}{p_H}\right)^{1/2} = \frac{w}{2}\lambda^{1/2}. \tag{81}$$

The ratio

$$\lambda = \rho L_H/p_H = 22.72, \tag{82}$$

is a number, yielding $l_a = 2.62$mm.

In principle, we still have to match this solution with the solution in the melting regime. However, for $V \to 0$ the melting regime shrinks to zero and the solution Eq. (79) applies to the whole region.

In the low velocity limit the expressions simplify, since the pressure in the layer approaches $p_H$. Going back to the first three expression (44)-(46), we have the equation for $l$

$$F_N = Mg = p_H wl, \qquad \text{or} \qquad l = \frac{F_N}{p_H w} \tag{83}$$

and $l$ becomes equal to the static contact length $2l_0$. The indentation depth $d$ approaches therefore $4\,d_0$, with $d_0$ the static value. The ploughing force reads in the limit $V \to 0$

$$F_{\text{pl}} = p_H wd = \frac{p_H wl^2}{2R} = \frac{F_N^2}{2p_H wR}, \tag{84}$$

using $l$ from Eq. (83). This is an interesting relation. At zero velocity there is no water layer and the ploughing force is the only friction. It shows that Amonton's law does not hold, since the friction is not proportional to the normal force. Note that the relation contains only the hardness $p_H$ and that it is therefore a relation to measure the hardness.

The integral for the friction due to the water layer becomes elementary

$$F_{\text{fr}} = \frac{\eta wV}{A}\int_0^l \frac{dx}{[\tanh((l-x)/\lambda)]^{1/2}} = p_H wk^{1/2}\lambda^{3/4}[l + l_a(0.5\log(2) + 0.25\pi)]. \tag{85}$$

As $k$ is proportional to $V$ the friction force vanishes as $V^{1/2}$.

# D   Heat transfer in the water layer

The heat flow $J$ in the water layer is related to the temperature $T$ by the equation

$$\mathbf{J} = -\kappa_{\text{w}}\nabla T \tag{86}$$

In the stationary state the divergence of $J$ equals the heat source density, which is given by Eq. (12)

$$\kappa_w \nabla^2 T = -\eta \frac{V^2}{h(x)^2}. \tag{87}$$

The solution of this equation has to be supplemented by the boundary conditions at the skate side $T = T_{sk}$ and the ice side $T = T_{ice}$. The main variation is parabolic in the downward $z$ direction. In terms of the coordinate $z'$ with respect to the center of the layer

$$z' = z - d(x) - h(x)/2, \tag{88}$$

we get the solution

$$T(z') = a + bz' - cz'^2. \tag{89}$$

The constant $c$ follows from Eq. (87) as

$$c = \frac{\eta}{2\kappa_w} \frac{V^2}{h(x)^2}. \tag{90}$$

The boundary conditions give the values of the constants $a$ and $b$.

$$\begin{cases} T_{sk} &= a - b\,h(x)/2 - c[h(x)/2]^2, \\ T_{ice} &= a + b\,h(x)/2 - c[h(x)/2]^2. \end{cases} \tag{91}$$

With $\Delta T = T_{ice} - T_{sk}$ we find for $b$

$$b = \frac{\Delta T}{h(x)}. \tag{92}$$

The unimportant parameter $a$ follows by using this value in one of the Eqs. (91).

With the temperature profile given we can determine the heat flows towards the skate and the ice. At the skate side we have a flow out of the water layer

$$J_{w-sk} = k_w[b + c\,h(x)] = \frac{k_w}{h(x)}[\Delta T + \Delta T_V], \tag{93}$$

where $\Delta T_V$ is a temperature difference depending only on the velocity $V$ and given by

$$\Delta T_V = \frac{\eta}{2\kappa_w} V^2 = 1.47 * 10^{-3} V^2. \tag{94}$$

(With $V$ the numerical value in m/s and $\Delta T_V$ in centigrade.) Likewise we have for the flow towards the ice the value

$$J_{w-ice} = -\kappa_w[b - c\,h(x)] = -\frac{\kappa_w}{h(x)}[\Delta T - \Delta T_V]. \tag{95}$$

The fraction $\zeta_w$ of the total heat produced in the layer towards the ice, is given by

$$\zeta_w = \frac{J_{w-ice}}{J_{w-sk}} = \frac{1}{2}\left(1 - \frac{\Delta T}{\Delta T_v}\right). \tag{96}$$

The temperature at the ice side equals the melting temperature at the pressure in the water layer. At the skate side the temperature may be higher than this melting temperature, but cannot be lower. So $\Delta T < 0$ and the fraction will always be higher than or equal to 1/2. If $-\Delta T > \Delta T_v$, all heat flows towards the ice.

We note that, due to the layer thickness $h(x)$ in the denominator of Eq. (95), the temperature gradient at the water-ice interface is huge.

# E Heat flows in the ice

The temperature distribution in the ice is governed by the heat equation

$$\frac{\partial T}{\partial t} = \alpha_{\text{ice}} \Delta T + \left( \frac{\partial T}{\partial t} \right)_{\text{forced}}. \tag{97}$$

First we have to find the expression for the temperature forcing. Take a point $x$ in the ice at time $t = 0$. This point has experienced for earlier times $t$ a temperature raise $T_m(x - Vt) - T_s$ at the surface, which we locate for convenience at $z = 0$. For the gradient in the $z$ direction this means a $\delta(z)$ dependence. So we find for the temperature forcing

$$\left( \frac{\partial^2 T(x,z,t)}{\partial z \partial t} \right)_{\text{forced}} = \frac{\partial [T_m(x - Vt) - T_s]}{\partial t} \delta(z) = -V \frac{\partial T_m(x - Vt)}{\partial x} \delta(z). \tag{98}$$

This holds for times in the past up to $t_0$

$$t_0 = -(l - x)/V, \tag{99}$$

with $l$ the contact length. Let us first discuss the case where the pressure at the tip has a melting temperature $T_m(l)$ *below* the surface temperature. Then we have to solve the following equation in the time interval $t_0 < t < 0$

$$\frac{\partial^2 T(x,z,t)}{\partial z \partial t} = \alpha_{\text{ice}} \Delta \frac{\partial T(x,z,t)}{\partial z} - 2V \frac{\partial T_m(x - Vt)}{\partial x} \delta(z). \tag{100}$$

We have inserted a factor 2 in the source term as it is easier to solve the equation in the complete space $-\infty < z < \infty$ and to use the symmetry between the upper and lower half $z$-plane. The differentiations in the Laplacian $\Delta$ may be restricted to those in the $z$ direction, since the variation in the $z$ direction is much larger than in the $x$ direction. The solution follows by Fourier transform in the $z$ direction

$$R_k(x,t) = \int_{-\infty}^{\infty} dz \frac{\partial T(x,z,t)}{\partial z} e^{ikz}. \tag{101}$$

The equation for $R_k(t)$ reads

$$\frac{\partial R_k(x,t)}{\partial t} = -\alpha_{\text{ice}} k^2 R_k(x,t) - 2V \frac{\partial T_m(x - Vt)}{\partial x}, \tag{102}$$

with the solution

$$R_k(x,t) = -2V \int_{t_0}^{0} dt' \frac{\partial T_m(x - Vt')}{\partial x} e^{\alpha_{\text{ice}} k^2 t'}. \tag{103}$$

The inverse Fourier transformation yields for the gradient at $z = 0$

$$\left( \frac{\partial T(x,z,t)}{\partial z} \right)_{z=0} = -V \int_{t_0}^{0} dt' \frac{1}{\sqrt{-\pi \alpha_{\text{ice}} t'}} \frac{\partial T_m(x - Vt')}{\partial x}. \tag{104}$$

Then changing the integration variable $t'$ to $x' = x - Vt'$ gives

$$\left( \frac{\partial T(x,z,t)}{\partial z} \right)_{z=0} = -\left( \frac{V}{\pi \alpha_{\text{ice}}} \right)^{1/2} \int_{x}^{l} dx' \frac{1}{\sqrt{x' - x}} \frac{\partial T_m(x')}{\partial x'}. \tag{105}$$

This expression holds for the case $T_m(l) < T_s$. In the other case, when $T_m(l) > T_s$, the ice temperature is suddenly raised at the tip by the amount $T_m(l) - T_s$ and one has in addition to the integral the contribution from this jump (leading to a $\delta$ function in the integral)

$$\delta\left(\frac{\partial T(x,z,t)}{\partial z}\right)_{z=0} = \left(\frac{V}{\pi\alpha_{\text{ice}}}\right)^{1/2}\frac{T_m(l) - T_s}{\sqrt{l-x}}. \tag{106}$$

The combination of the integral and the jump are given in Eq. (53). In [8] only this jump is taken into account with $T_m(l) = 0$.

Here we give for completeness the change in the layer equation as due to this jump, in order to show how the layer equation of Lozowski and Szilder [8] results. The fraction of heat available for melting is then reduced by the factor

$$\zeta = 1 - \frac{2\kappa_{\text{ice}}[T_m(l) - T_s]h(x)}{\eta V^{3/2}\sqrt{\pi\alpha_i(l-x)}}. \tag{107}$$

We find $\zeta(\bar{x})$ by scaling $h(x)$ and $l - x$

$$\zeta(\bar{x}) = 1 - q\frac{\bar{h}(\bar{x})}{[\bar{l} - \bar{x}]^{1/2}}, \tag{108}$$

with the constant

$$q = \frac{\sqrt{2}\kappa_{\text{ice}}}{\sqrt{\eta\pi\alpha_{\text{ice}}\rho L_h}}\frac{T_m(l) - T_s}{V} = 1.825\frac{T_m(l) - T_s}{V}. \tag{109}$$

The correction due to $\zeta$ changes the scaled equation (34) to

$$-\frac{d\bar{h}}{d\bar{x}} = \frac{1}{\bar{h}(\bar{x})} - \frac{q}{\sqrt{\bar{l} - \bar{x}}} - \frac{\bar{x} + c_1}{c_2 + \bar{h}^3(\bar{x})}\bar{h}^3(\bar{x}), \tag{110}$$

Eq. (110) is the scaled version of a similar equation for the layer given in [8].

It is interesting that $\zeta$ in Eq. (108) approaches at the tip a finite value $\zeta(\bar{l})$, since $\bar{h}(\bar{l} - \bar{x})$ vanishes in the same way as the square root

$$\bar{h}(\bar{l} - \bar{x}) \simeq a\sqrt{\bar{l} - \bar{x}}. \tag{111}$$

The amplitude $a$ satisfies the equation

$$\frac{a}{2} = \frac{1}{a} - q, \qquad \text{or} \qquad a = \sqrt{2 + q^2} - q. \tag{112}$$

For $q \to 0$ the amplitude $a = \sqrt{2}$ (as before in Eq. (22)) and for $q$ large, the amplitude vanishes as $1/q$. So for low temperatures and slow velocities the value of $a$ rapidly decreases, rendering the thickness of the water layer too thin to treat the layer as a hydrodynamic system.

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
