# Peer review of "Skating on slippery ice"

_SciPost Physics, doi:SciPost Phys. 3, 042 (2017)_

## Round 2 · Referee Report · Anonymous (Referee 1) · 2017-8-10

Strengths
Gives very reasonable results.
Very interesting topic.
Weaknesses
Only very little comparison to experiment.
Bite Angle under which the skate contacts the ice surface is ignored.
Thermal properties of the skate are ignored.
Report
As such, the manuscript forms a significant improvement to existing ice skating theories. I therefore recommend the manuscript to be published if the following changes/questions are satisfactorily addressed by the author:
Requested changes
-
If the skate on ice contact is modelled as a Hertzian contact between a cylinder with radius 22 m, length 1.1 mm and a second cylinder with equal length and infinite radius, the contact pressure does not exceed 3 MPa which should be below the penetration hardness of the ice. Should the conclusion not be that the contact is elastic?
-
The angle between the skate and the ice (bite angle) is assumed to be 0. The bite angle will impact, for instance, the stationary length of contact, $l_0$, strongly. The experiments from de Koning et al suggest that the friction force is not too sensitive to the bite angle. Is this also expected for the present theory? I suppose that if the water layer can fill the gap between skate and ice caused by the bite angle the effect on friction will be limited?
-
Page 4 below eq. 4: perhaps 'in contact with the skate' instead of 'in touch with the skate'.
-
Page 6 below eq. 8 'mayor' should be 'major'
-
I do not understand equation 9: I would expect the speed with which the ice recedes, $v_{ice}$, to be the sum of the speed with which the skate recedes, $v_{sk}$, and the speed with which the water layer grows,$v_m$.
-
Where does the speed, $V$, in equation 10 come from?
-
Why is the thermal conductivity of the skate no parameter in the theory? I find it surprising that more heat flows into the ice than into the skate although the skate can in principle transport heat faster.
-
It may be good to mention that the viscosity of water is not very sensitive to the typical ice skating pressures.
-
page 8 above eq. 21: 'Eq. (21) dominates' should be 'Eq. (20) dominates'
-
For the calculation of the friction force that results from shearing the water film; does the -l<x<0 part of the skate not also shear a water film?
-
If the heat source for melting ice is sheared water, where does that water come from (it is not present at V=0)?
-
The data from reference 13 (FIG 3) should allow for an experimental estimate of $\gamma$ and show whether or not the Bingham model is appropriate for ice. It may also be interesting to compare the penetration hardness from ref 13 to that reported in the paper.
-
Page 16 bottom 'treat it hydrodynamic' should be 'treat it as a hydrodynamic system'
-
Bottom page 17 'not very emcouraging' should be 'not very encouraging'
-
Appendix A under eq. 58 it says 'skate blade z = h(x)' this should be d(x)
-
Appendix C refers to expressions 66-68 above equation 82. Should this be 7?

---

## Round 2 · Referee Report · Anonymous (Referee 2) · 2017-8-21

Strengths
- Detailed treatment based on hydrodynamic approach for clarifying the origin of slippery ice surface
- Combining the theory of Le Berre and Pomeau [9], which only accounts for the e ects of melting and the theory of Lozowski and Szilder, which equals the pressure in the water layer to the hardness of ice to overcome the phenomena of slippery ice surface
Weaknesses
- The assumptions the author uses are bit far from the knowledge obtained from microscopic measurement and molecular dynamics simulation
Report
As is written as the strength of the papers, I enjoy reading the manuscript, as it provides insight into the long-lived question - why the ice surface slippery - based on the hydrodynamics approaches. The analysis is consistent with the well-known phenomena, meaning that the theory presented here could be a solid hypothesis. However, I am not convinced whether this theory can explain the ice friction behavior beyond the skating condition. It seems very important to clarify in which conditions the current theory is valid and in which condition the assumption is broken down. We can control the weight and velocity as well as temperature in a much wider range than the ranges discussed here. It is important to comment on how this theory is applicable for such wider ranges of the physical condition.
For the microscopic measurement on the ice surface - in particular the liquid water on the top of the ice surface, I would suggest the author to read several literature. - http://pubs.acs.org/doi/abs/10.1021/acs.chemrev.6b00045 (Fig. 16) - http://aip.scitation.org/doi/abs/10.1063/1.2940195 These may be contradictory with the author's assumption, but it seems quite important to comment on the consistency/inconsistency of the author's assumption with these measurement/computational analysis.
Requested changes
- In Figure 8, the titles of the axes should be given.

---

## Round 3 · Referee Report · Anonymous · 2017-10-12

Report

I thank the author for clarifying the points that were raised and recommend the paper to be published.

---

## Round 3 · Author Response

The constructive remarks of the referees are highly appreciated. With respect to the general points raised by the referees, I would like to reply the following.

  • Comparison with experiments. Indeed there is regrettably little comparison with experiments. This is mainly due to the fact that few relevant experiments exist. The investigation started as an attempt to explain direct measurements of the thickness of the water layer. However the laboratory experiments could not reliably detect water layers. This conforms the theoretical calculations leading to water layers of the order of fractions of a mu m thick. Such water layers are, so far, too thin to see in the variation of the dielectric constant.

-What if the water layer is absent? Friction generates heat, no matter the origin of the friction. Whether the heat leads to melting depends on the temperature of the ice. We find in Section 13 that at low temperatures and slow velocities, the heat available for melting becomes so small, that the water layer thickness becomes sub-hydrodynamical. Under these circumstances our calculation is not applicable. For velocities relevant for skating the water layer can be treated as a hydrodynamical system and the friction is then given by the friction in the water layer and independent of the surface properties of ice and skate.

If the formation of the water layer plays no role, the friction is determined by the properties of the surface of ice. This is demonstrated in measurements of B. Weber et al. (ref. 5) where a friction is found orders of magnitude higher than for skating conditions.

-Role of the bite angle. The calculation is restricted to a horizontal skate, while a finite bite angle is indeed important for real skating, since it can not be avoided. It is likely that a finite bite angle can be discussed in the same context as the present paper (see the work of Le Berre and Pomeau, ref.~9), but it would have made the already too-long paper even longer, without adding to the essential mechanism of skating.

-Thermal properties of the skates. The thermal properties of the skate are not ignored. In the appendix D, the heat flow inside the water layer is discussed, which is determined by the temperatures of ice surface and the skate. Since the skate will not be colder than the ice, the fraction of heat flowing towards the skate will not be larger than 50% of the produced heat. This is independent of the thermal properties of the skate.

-Validity of the assumptions. I could not find any direct experimental arguments pro or contra the basic assumption Eq.~(6). Thus the proof or disproof of (6) has to come from measurements. It requires firstly a precision measurement of the hardness under quasi-static intrusion in order to determine the excess pressure. Next it requires intrusion rates under controlled speed and pressure. To the best of my knowledge these experiments, comparable to the intrusion rate of skates, are not available.

I have made in the introduction of the paper a number of modifications to illuminate the above mentioned points. A number of improvements suggested by the first referee are taken over without change i.e. the remarks 3,4,8,9,13,14,15 and 16. I am grateful for the scrutiny of this referee.

The other points give rise the following commentary.

Point 1. The calculation of the Hertzian contact between the skate, seen as a cylinder of curvature 22 m and length 1.1 mm, and an equal-length ice-cylinder of infinite radius, is not relevant for the question whether the deformation of a skate is elastic or plastic. For that question the deformation sideways from the skate is most important. In particular, assuming that the skate has sharp edges, the deformation will always be plastic, since a sharp edge yields a kink in the ice surface, leading to a diverging counter pressure. In practise the edges are rounded off and that makes the question elastic vs plastic ill-posed. The counter pressure is maximal at the edges and therefore the degree of rounding off determines the limit of elastic deformations

Point 5. The rate of change of the through made by the skate is due to melting and ploughing. So it is the sum of the melting velocity v_m and the downward ploughing (or receding velocity) v_{ice}. I have made this more explicit in the text.

Point 6. The velocity comes in as the ratio of the derivative with respect to position and to time.

Point 10. What happens with the water layer, after the lowest point of the skate has passed, is irrelevant for the friction with the skate, since there is no more contact with the skate. I have stated this explicitly in the revised text.

Point 11. The water layer originates in melting. For V=0 no water is generated and only ploughing remains in the theory. The ploughing force is given by the hardness, provided the hardness is measured quasi-statically. This would indeed yield a method to measure the hardness of ice.

Fig. 8. In the program that generates Fig. 8, it is hard to give the axis appropriate labels. The role of the axes is clarified in the caption.

Hopefully the changes made according to this reply have improved the paper.

You are currently on this page

Resubmission 1706.08278v3 on 22 September 2017

---

## Editorial Decision

published